# Synthesis and Behavior of DNA Oligomers Containing the Ambiguous Z-Nucleobase 5-Aminoimidazole-4-carboxamide

**DOI:** 10.3390/molecules28073265

**Published:** 2023-04-06

**Authors:** Yuhei Nogi, Noriko Saito-Tarashima, Sangita Karanjit, Noriaki Minakawa

**Affiliations:** Graduate School of Pharmaceutical Science, Tokushima University, 1-78-1 Shomachi, Tokushima 770-8505, Japankaranjit@tokushima-u.ac.jp (S.K.)

**Keywords:** chemically modified oligonucleotide, 5-aminoimidazole-4-carboxamide, single nucleotide insertion, base-pairing property, melting temperature, mutagenic nucleosides

## Abstract

5-Amino-1-β-D-ribofuranosylimidazole-4-carboxamide 5′-monophosphate (ZMP) is a central intermediate in de novo purine nucleotide biosynthesis. Its nucleobase moiety, 5-aminoimidazole-4-carboxamide (Z-base), is considered an ambiguous base that can pair with any canonical base owing to the rotatable nature of its 5-carboxamide group. This idea of ambiguous base pairing due to free rotation of the carboxamide has been applied to designing mutagenic antiviral nucleosides, such as ribavirin and T-705. However, the ambiguous base-pairing ability of Z-base has not been elucidated, because the synthesis of Z-base-containing oligomers is problematic. Herein, we propose a practical method for the synthesis of Z-base-containing DNA oligomers based on the ring-opening reaction of an *N*^1^-dinitrophenylhypoxanthine (Hxa^DNP^) base. Thermal denaturation studies of the resulting oligomers revealed that the Z-base behaves physiologically as an A-like nucleobase, preferentially forming pairs with T. We tested the behavior of Z-base-containing DNA oligomers in enzyme-catalyzed reactions: in single nucleotide insertion, Klenow fragment DNA polymerase recognized Z-base as an A-like analog and incorporated dTTP as a complementary nucleotide to Z-base in the DNA template; in PCR amplification, Taq DNA polymerase similarly incorporated dTTP as a complementary nucleotide to Z-base. Our findings will contribute to the development of new mutagenic antiviral nucleoside analogs.

## 1. Introduction

5-Amino-1-β-D-ribofuranosylimidazole-4-carboxamide 5′-monophosphate (ZMP, also known as AICAR 5′-monophosphate) is a central intermediate in the de novo purine nucleotide biosynthesis, an ancient metabolic pathway common to most organisms. In the final step of purine nucleotide biosynthesis, ZMP is converted into 5-formylated ZMP, which in turn is converted into inosine 5′-monophosphate, a precursor of AMP and GMP.

The nucleobase of ZMP, 5-aminoimidazole-4-carboxamide (Z-base), is expected to pair with any canonical base, owing to the rotatable nature of its 5-carboxamide group (Figure 1a) [1,2]. This flexibility of the carboxamide group has been utilized in the design of mutagenic antiviral nucleosides [3,4]. For example, ribavirin (1-β-D-ribofuranosyl-1,2,4-triazole-3-carboxamide) has a rotatable 3-carboxamide group on its triazole nucleobase (Figure 1b) [5,6,7], and in vitro studies have shown that the hepatitis C virus (HCV) RNA-dependent RNA polymerase (RdRp) can incorporate ribavirin 5′-triphosphate as a nucleotide analog of either ATP or GTP [8]. The ambiguous behavior of ribavirin causes G-to-A and C-to-U transition mutations in the viral genome and results in antiviral activity against a variety of RNA viruses [7,9,10,11]. Furthermore, when ribavirin itself is used as the template sequence, ribavirin-templated incorporation of CTP by HCV RdRp is four times more efficient than that of UTP, indicating that ribavirin, as an ambiguous purine analog, shows bias towards being more guanine (G)-like than adenine (A)-like.

Similarly, T-705 (6-fluoro-3-hydroxy-2-pyrazinecarboxamide, favipiravir), a broad-spectrum antiviral agent clinically used to treat influenza infection, is also an ambiguous base-pairing analog (Figure 1c) [12,13,14]. Due to free rotation of the 2-carboxamide group of T-705, the viral polymerase recognizes T-705 ribonucleoside 5′-triphosphate (T705-TP) as both a G-like and an A-like analog [15]. Detailed kinetic analysis of the single nucleotide insertion reaction has shown that T705-TP is more efficiently incorporated as a complement to C rather than U on the viral RNA template; which, means that the 2-carboxamide group of T-705 is preferentially present in a G-like configuration.

For Z-nucleotides, by contrast, there are few reports on whether the Z-base behaves as an ambiguous base, except for the study of Sabina et al., which shows that ZTP, as well as ATP, is a substrate for the reverse reaction catalyzed by 5-phosphoribosyl-1-pyrophosphate (PRPP) synthase [16].

Herein, therefore, we have explored a simple method to prepare DNA oligomers containing Z-bases based on the ring-opening reaction of an *N*^1^-dinitrophenylhypoxanthine (Hxa^DNP^) base. By performing thermal and thermodynamic analysis of the resulting DNA duplexes containing a Z-base, we reveal that Z-base behaves as an A-like analog that forms a stable pair with T, rather than a G-like analog that forms a pair with C. Furthermore, we show that DNA polymerases catalyze Z-templated incorporation of TTP, but little or no CTP. Our results suggest that the ambiguous Z-base functions as an A-like analog in DNA duplexes.

## 2. Results and Discussions

### 2.1. Chemistry

To prepare DNA oligomers containing Z-bases, it is important to protect the Z-base. As we and others have previously reported, the sensitivity of the Z-base to ring closure into a purine, coupled with the possible dehydration of the 4-carboxamide group to form a cyano group, renders the choice of protecting group for the Z-base more difficult relative to standard nucleobases [1,17]. In previous studies, therefore, construction of Z-base by heat treatment of a *N*^1^-dinitrophenylhypoxanthine (Hxa^DNP^) base with ethylenediamine has been applied to the synthesis of various Z-nucleoside or Z-nucleotide analogs [17,18,19,20,21]. In this study, to prepare DNA oligomers containing Z-base via this approach, the 5′-OH group of 2′-deoxyinosine (**1**) was first protected with 2,4-dimethoxytrityl chloride (DMTrCl) in pyridine with 75% yield (Figure 1). The resulting product **2** was treated with 1-chloro-2,4-dinitrobenzene in DMF in the presence of K_2_CO_3_ to give **3** (85% yield). Lastly, **3** was converted to the phosphoramidite derivative **4** (66%) under standard conditions.

With the desired phosphoramidite unit **4** in hand, we next prepared different DNA oligomers containing Z-base. The schematic procedure is shown in Figure 2a. First, a 15-mer oligodeoxynucleotide (ODN) with a Hxa^DNP^-base in the middle of the sequence (**ODN1**) was prepared by using an automatic DNA/RNA synthesizer. It was then treated with ethylenediamine at 50 °C for 30 min, and an aliquot of the reaction mixture was analyzed by using LC-MS. The spectrum showed a major peak corresponding to the desired **ODN2** possessing a Z-base (RT = 7.9 min, calcd for C_145_H_186_N_54_O_88_P_14_ 4526.9700, observed 4526.9759), implying that (i) ring-opening of the Hxa^DNP^-base to form Z-base, (ii) nucleobase deprotection of canonical bases [*N*^6^-benzoyl (Bz) group for adenine, *N*^2^-isobutyryl group for guanine, and *N*^4^-acetyl group for cytosine], and (iii) cleavage from the resin had successfully proceeded in one step (Figure 2b) [22]. However, a minor impurity peak (RT = 7.6 min) was also observed and determined to be **ODN3** containing an *N*^1^-alkylated Hxa base. We also examined the potential incorporation of multiple Z-bases in **ODN4**; however, the undesired formation of *N*^1^-alkylated Hxa bases resulted in complex mixtures (Appendix A: Appendix A).

Our results allowed us to propose the following reaction mechanism (Appendix A). When the Hxa^DNP^-base **S1** is treated with ethylenediamine at 50 °C, the nucleophilic attack of the ethylenediamine on C2 of hypoxanthin ring occurs to form formamidine intermediate **S2**, which then undergoes the attack of a second diamine on the dinitrophenyl ipso carbon to yield the **S3**. Subsequent intramolecular attack on the formamidine carbon by the free terminal amino group of **S3** leads to formation of the cyclic orthoamide **S4** (path A). Then, a third diamine attacks the cyclic orthoamide carbon to yield the desired Z-base **S5**. However, if the N atom linked to the amidine carbon in **S3** participates in reclosure of the purine by attacking the carbonyl group, the undesired *N*^1^-alkylated Hxa derivative **S6** will form (path B). 

To prove our hypothesis, we used density functional theory (DFT) to calculate thermodynamic parameters for the model substrate **S1** with the sugar moiety replaced with a methyl substituent. We started by investigating the reaction profile from the intermediate **S2**, which is obtained via the ring opening reaction of **S1** with ethelenediamine (*n* = 2). The calculations showed that, formation of the five-membered cyclic orthoamide (**S3** → **S4** via **TS_S3-S4_**, 40.9 kcal/mol) is the rate-limiting step to give the desired **S5** (path A). In addition, the by-product **S6** can form via path B (**TS_S3–S6_**, 38.7 kcal/mol). However, when the activation energies were calculated for the path A and path B taking the larger diamine (*n* = 3), the outcome was opposite (path A 38.2 kcal/mol versus path B 41.4 kcal/mol). This result of DFT calculations for our proposed mechanism suggested that the construction of a six-membered cyclic orthoamide with a longer C-chain diamine would proceed more easily than a five-membered one. Thus, we explored the Z-base construction with 1,3-propanediamine (*n* = 3), which would form a six-membered cyclic orthoamide intermediate.

Figure 2c shows the result of the treatment of support-bonded **ODN1** with 1,3-propanediamine at 50 °C. Under these conditions, formation of the undesired *N*^1^-alkylated Hxa product was markedly reduced and the isolated yield of the desired **ODN2** containing Z-base after HPLC purification was 35%; which, is similar to that in normal ODN synthesis. Moreover, this synthesis of **ODN4** containing multiple Z-bases gave the desired sequence as the main product (Appendix A).

### 2.2. Physical Assessment of the Base-Pairing Ability of Z-Base

Next, we investigated the base-pairing ability of Z-base by evaluating the melting temperature (*T*_m_) of duplexes by ultraviolet (UV) absorbance. The *T*_m_ of 15-mer duplexes containing Z-base was measured in a buffer comprising 100 mM NaCl, 10 mM Na_2_HPO_4_, and 1 mM Na_2_EDTA (pH 7.0) (Figure 3). When a Z:T pair was formed at a position sandwiched between A:T pairs in the DNA duplex, the *T*_m_ value was 53.9 °C (entry 1, Figure 3). By contrast, the *T*_m_ value of a Z:C pair was much lower (entry 2, 43.8 °C) and almost equal to that of the natural mismatched A:C and A:A base pairs (entries 7 and 8, 47.7 °C and 49.4 °C, respectively). These results indicate that Z-base behaves as an A-like analog, forming a stable base pair with T in the DNA duplex, rather than as a G-like analog, which pairs with C. Furthermore, the possible formation of Z:G pairs (*T*_m_ = 52.4 °C, entry 4) strongly supports the idea that Z-base functions as an A-like analog, because it suggests the formation of a Hoogsteen-type base pair similar to A:G pairs (*T*_m_ = 51.6 °C, entry 9) (Appendix A). The calculated thermodynamic parameters (Δ*H*°, Δ*S*°, and Δ*G*°) also support these considerations. Given the influence of adjacent base pairs, we also evaluated the *T*_m_ values of duplexes in which the Z-base was sandwiched between G:C pairs (entries 10–13); which, showed that Z-base displayed similar base-paring behavior as when it was sandwiched between A:T pairs (entries 1–4).

The duplex containing a Z:T pair was less stable than the fully complementary A:T and C:G duplexes (entry 1 versus entries 5 and 6, Figure 3). We considered that this destabilization is due to loss of the stacking affinities of the nucleobase, which has been altered to an imidazole ring (Z-base) from a purine ring (A- and G-bases). Next, therefore, we prepared **ODN6** with either a Z-base or a natural nucleobase at the position of the dangling end and evaluated the *T*_m_ values and thermodynamic parameters of the homoduplexes according to the method of Kool et al. [23] (Figure 4). The results showed that the ΔΔ*G*° stacking affinity of a Z-base (imidazole) (entry 3) was less favorable than that of an A-base (purine) (entry 2) (–2.4 kcal mol^–1^ versus –1.2 kcal mol^–1^). Therefore, a decrease in the stacking affinity is likely to be one of the main reasons for the reduced thermal and thermodynamic stability of a Z:T pair relative to a natural A:T pair.

### 2.3. Enzymatic Assessment of the Base-Pairing Ability of Z-Base

Next, we assessed the behavior of the Z-base in enzyme-mediated processes by first examining single-nucleotide insertion by Klenow fragment DNA polymerases. The experiments were carried out with a template strand containing Z-base at position 21 from the 3′-end (represented as X, 30-mer, **ODN 7**), and a 5′-fluoresceinisothiocyanate (FITC)-labeled primer (20-mer) (Figure 5a) in the presence of each 2′-deoxynucleoside 5′-triphosphate (dNTP). As shown in Figure 5b, with the 3′→5′ exonuclease-deficient Klenow fragment (KF exo^–^), only dTTP was incorporated as a complementary base against the Z-base in the template, giving a 21-mer sequence (lane 8) just as when the natural A-base was included in the template (lane 4); whereas, no other dNTPs were incorporated (lanes 6, 7 and 9). Moreover, the Klenow fragment with 3′→5′ exonuclease activity (KF exo^+^) also formed Z:T as a complementary base pair (lane 17), indicating that the Z:T pair was recognized as a natural Watson–Crick-like matched base pair.

To quantify the selectivity and efficiency of the single-nucleotide insertion experiments, we determined kinetic parameters, including the Michaelis constant (*K*_m_), maximum rate of enzyme reaction (*V*_max_), and incorporation efficiency (*V*_max_/*K*_m_), for the reaction by using each dNTP at various concentrations (Figure 5c). We found that the efficiency of dTTP incorporation was almost 19 times lower against Z-base in the template than against A in the template (Z:T pair, *V*_max_/*K*_m_ = 1.37 × 10^7^; A:T pair, 2.62 × 10^8^). Nevertheless, the parameters strongly suggested that the ambiguous Z-base functions as an A-like analog during replication by DNA polymerase, forming a pair with T.

We also explored PCR amplification catalyzed Taq DNA polymerase with a DNA template containing Z-base. The experiment was carried out with an 87-mer template containing Z-base and two primers to give 104-bp amplicons (Figure 5d), and the resulting amplicons were sequenced to evaluate accumulated mutagenesis that occurred at the position of Z-base. As shown in Figure 5e, the PCR for the template with Z-base in the middle of the sequence proceeded with efficiency comparable to that of the natural template, yielding nearly equal amounts of amplicons. In addition, sequencing of the amplified product verified that dTTP was incorporated against Z-base in the template as a complementary nucleotide (Figure 5f). As described in the Introduction, the rotatable carboxamide group on ribavirin and T-705 prefers the G-like configuration rather than the A-like configuration. In contrast to ribavirin and T-705, our current results indicate that Z-base acts as an A-like analog. We consider that this different behavior might be influenced by the group neighboring the carboxamide in each nucleobase. Thus, in the case of ribavirin, a hydrogen bonding interaction between the N2 atom and the 3-carboxamide hydrogen atom would give preference to the G-like configuration (Figure 6a). Similarly, a potential interaction between the O3 atom and the 2-carboxamide hydrogen atom of T-705 would stabilize the G-like configuration (Figure 6b). By contrast, the 4-carboxamide group of Z-base prefers the A-like configuration due to a stabilizing interaction between the 5-amino hydrogen and the 4-carboxamide group (the G-like configuration would cause steric repulsion, Figure 6c). We believe that our results will contribute to the molecular design of new mutagenic antiviral nucleoside analogs.

## 3. Conclusions

In this study, we have succeeded in establishing a practical synthesis of DNA oligomers containing Z-base. Based on our precise MS analysis and DFT calculation-based considerations, we found that 1,3-propanediamine treatment was more effective than ethylenediamine treatment for the construction of Z-base via the ring-opening reaction of Hxa^DNP^-base under heating conditions. Thermal and thermodynamic denaturing studies of the resulting DNA oligomers revealed that Z-bases preferentially form Z:T pairs like natural A:T pairs with the possible formation of a Hoogsteen-type Z:G pair similar to an A:G pair. Collectively, our results indicate that Z-base prefers an A-like configuration in the DNA duplex. To understand how Z-base behaves in enzyme-catalyzed processes, we measured the products of single nucleotide insertion by Klenow fragment DNA polymerase with a DNA template containing a Z-base in its sequence. With or without proofreading 3′→5′ exonuclease activity, Klenow fragment DNA polymerase selectively incorporated dTTP as a nucleotide complementary to the Z-base in the template, forming Z:T base pairs. PCR amplification of a template containing a Z-base also showed convergence to A:T pairs. 

The behavior of Z-base as an A-like analog, as revealed by our results, distinguishes it from other mutagenic nucleosides with a carboxamide group on the nucleobase such as ribavirin and T-705, which both prefer a G-like configuration. We believe that this finding will contribute to the development of new mutagenic antiviral nucleoside analogs. Further evaluation of Z-base in transcription is currently underway.

## 4. Materials and Methods

### 4.1. General Methods

Analytical TLC was performed on Kieselgel F_254_ plates (Merck, Darmstadt, Germany) and visualized by UV light (254 nm, Axel, Osaka, Japan). Column chromatography was performed using Chemical silica gel 60N (neutral, KANTO Chemical, Tokyo, Japan). Physical data were obtained as follows. NMR spectra were recorded on a Bruker FT-NMR AV400 or AV500 instrument (Billerica, MA, USA). ^1^H NMR spectra were recorded at 400 or 500 MHz, referenced to TMS in CDCl_3_ (0.00 ppm), DMSO-*d*_6_ (2.50 ppm). ^13^C NMR spectra were recorded at 100 or 125 MHz, referenced to TMS in CDCl_3_ (0.00 ppm). ^31^P NMR spectrum was recorded at 162 or 202 MHz, referenced to phosphoric acid as an external reference in CDCl_3_ (0.00 ppm). Chemical shifts are reported in parts per million (δ), and signals are expressed as s (singlet), d (doublet), t (triplet), q (quartet), m (multiplet), or br (broad). All exchangeable protons were detected by addition of D_2_O. Mass spectra were recorded on a quadrupole SQD2 spectrometer (Waters, Milford, MA, USA) for LRMS or a TOF SYNAPT G2-Si HDMS spectrometer (Waters) for HRMS. All the chemical reagents and solvents used were commercially available and used without further purification.

### 4.2. Chemistry

**5′-*O*-(4,4′-Dimethoxytrityl)-2′-deoxyinosine (2).** Under an argon atmosphere, to a solution of **1** (2.72 g, 10.8 mmol) in pyridine (100 mL) was added DMTrCl (4.39 g, 13.0 mmol) at 0 °C, and the whole was stirred for 7 h at room temperature. The reaction was quenched by addition of ice and concentrated in vacuo. Next, AcOEt was added to the residue and the resulting white precipitate was collected by filtration, and then washed with H_2_O followed by AcOEt to give **2** (4.26 g, 71%) as a white powder. LRMS (ESI) *m*/*z*: [M + Na]^+^ 577; HRMS (ESI) *m*/*z*: [M + H]^+^ calcd for C_31_H_31_N_4_O_6_ 555.2238; found 555.2282; ^1^H NMR (DMSO-*d*_6_, 400 MHz) δ 12.38 (1 H, d, *J* = 3.9 Hz, exchangeable with D_2_O, -NH), 8.19 (1 H, s, H-8), 7.99 (1 H, d, *J* = 3.9 Hz, H-2), 7.33–7.31 (2 H, m, DMTr), 7.25–7.18 (7 H, m, DMTr), 6.83–6.78 (4 H, m, DMTr), 6.34 (1 H, dd, *J* = 6.3, 6.3 Hz, H-1′), 5.38 (1 H, s, exchangeable with D_2_O, -OH), 4.44–4.40 (1 H, m, H-3′), 3.99–3.95 (1 H, m, H-4′), 3.72 (each 3 H, each s, OMe), 3.17 (1 H, dd, *J* = 6.3, 10.2 Hz, H-5′a), 3.12 (1 H, dd, *J* = 3.9, 10.2 Hz, H-5′b), 2.77 (1 H, ddd, *J* = 6.3, 6.3, 12.9 Hz, H-2′a), 2.34 (1 H, ddd, *J* = 4.8, 6.3, 12.9 Hz, H-2′b); ^13^C{^1^H} NMR (DMSO-*d*_6_, 100 MHz) δ 158.0, 158.0, 156.6, 147.9, 145.6, 144.9, 138.7, 135.6, 135.5, 129.7, 129.6, 127.7, 127.7, 126.6, 124.6, 113.1, 113.1, 85.9, 85.4, 83.5, 70.5, 64.1, 55.0, 55.0.

***N*^1^-(2,4-Dinitrophenyl)-5′-*O*-(4,4′-dimethoxytrityl)-2′-deoxyinosine (3).** Under an argon atmosphere, to a solution of **2** (1.66 g, 3.0 mmol) in DMF (30 mL) containing K_2_CO_3_ (829 mg, 6.0 mmol) was added 1-chloro-2,4-dinitrobenzene (1.22 g, 6.0 mmol), and the mixture was heated for 1 h at 80 °C in an oil bath. After being cooled to room temperature, the reaction mixture was filtered through a celite pad and the residue was washed with AcOEt. The solvent was removed in vacuo and the residue was dissolved in AcOEt. The organic layer was washed with H_2_O (three times) followed by brine. The separated organic layer was dried (Na_2_SO_4_) and concentrated in vacuo. The residue was purified by a silica gel column, eluted with MeOH in CHCl_3_ (0–6%), to give **3** (1.85 g, 85%) as a yellow foam. LRMS (ESI) *m*/*z*: [M + Na]^+^ 743; HRMS (ESI) *m*/*z*: [M + H]^+^ calcd for C_37_H_33_N_6_O_10_ 721.2253; found 721.2254; ^1^H NMR (CDCl_3_, 400 MHz) δ 9.03 (each 0.5 H, each d, *J* = 2.8 Hz, DNP), 8.65 (0.5 H, dd, *J* = 2.4, 2.4 Hz, DNP), 8.62 (0.5 H, dd, *J* = 2.4, 2.4 Hz, DNP), 7.98 (0.5 H, s, H-8), 7.96 (0.5 H, s, H-8), 7.93 (0.5 H, s, H-2), 7.85 (0.5 H, s, H-2), 7.66 (0.5 H, d, *J* = 8.6 Hz, DNP), 7.61 (0.5 H, d, *J* = 8.6 Hz, DNP), 7.41–7.39 (2 H, m, DMTr), 7.31–7.19 (7 H, m, DMTr), 6.83–6.79 (4 H, m, DMTr), 6.44 (0.5 H, dd, *J* = 6.5, 6.5 Hz, H-1′), 6.40 (0.5 H, dd, *J* = 6.5, 6.5 Hz, H-1′), 4.72–4.66 (1 H, m, H-3′), 4.18–4.15 (1 H, m, H-4′), 3.77 (each 1.5 H, each s, OMe), 3.77 (3 H, 2 s, OMe), 3.47, 3.44 (1 H, each dd, *J* = 4.6, 10.2 Hz, *J* = 4.4, 10.2 Hz, H-5′a), 3.37, 3.36 (1 H, each dd, *J* = 3.8, 10.6 Hz, J = 3.9, 10.2 Hz, H-5′b), 2.84 (0.5H, ddd, *J* = 6.5, 6.5, 13.3 Hz, H-2′a), 2.72 (0.5 H, ddd, *J* = 6.5, 6.5, 13.3 Hz, H-2′b), 2.60 (0.5 H, ddd, *J* = 4.5, 6.5, 14.1 Hz, H-2′a), 2.57 (0.5 H, each ddd, *J* = 4.5, 6.5, 13.3 Hz, H-2′b), 2.28 (0.5 H, d, *J* = 3.5 Hz, exchangeable with D_2_O, -OH), 2.23 (0.5 H, d, *J* = 3.5 Hz, exchangeable with D_2_O, -OH); ^13^C{^1^H} NMR (CDCl_3_, 125 MHz) δ 158.6, 158.6, 158.6, 155.2, 148.1, 147.3, 147.2, 146.3, 146.3, 144.9, 144.7, 144.5, 144.5, 139.5, 138.9, 135.7, 135.6, 135.6, 135.6, 135.5, 131.9, 131.8, 130.1, 130.1, 130.0, 128.8, 128.8, 128.1, 128.1, 128.0, 127.9, 127.0, 127.0, 124.4, 124.1, 121.3, 121.3, 113.3, 113.2, 86.7, 86.6, 86.4, 86.3, 84.8, 84.3, 72.4, 72.3, 63.8, 63.7, 55.3, 40.9, 40.4.

***N*^1^-(2,4-Dinitrophenyl)-3′-*O*-{2-cyanoethoxy-(*N*,*N*-diisopropylamino)phosphino}-5′-*O*-(4,4′-dimethoxytrityl)-2′-deoxyinosine (4).** Under an argon atmosphere, a solution of **3** (360 mg, 0.5 mmol) in CH_2_Cl_2_ (10 mL) containing 4Å MS (200 mg) was stirred for 1 h at room temperature. Then, 2-cyanoethyl-*N*,*N*-diisopropylchlorophosphoramidite (167 mL, 0.75 mmol), *N*,*N*-diisopropylethylamine (DIPEA) (348 mL, 2.0 mmol), and 4-dimethylaminopyridine (DMAP) (3 mg 0.025 mmol) were added to the above solution at 0 °C, and the whole was stirred for 80 min at room temperature. The reaction mixture was diluted with CHCl_3_, washed with H_2_O (twice), and saturated NaHCO_3_ followed by brine, dried over Na_2_SO_4_, and concentrated in vacuo. The residue was purified by a silica gel column, eluted with hexane/AcOEt (1/2–1/3), to give **4** (304 mg, 66%) as a pale brown foam. LRMS (ESI) *m*/*z*: [M + Na]^+^ 943; HRMS (ESI) *m*/*z*: [M + H]^+^ calcd for C_46_H_50_N_8_O_11_P 921.3331; found 921.3375; 1H NMR (CDCl_3_, 500 MHz) δ 9.05 (0.5 H, d, *J* = 2.5 Hz, DNP), 9.04 (0.5 H, d, *J* = 2.5, DNP), 8.67–8.63 (1 H, m, DNP), 8.04 (0.2 H, s, H-8), 8.01 (0.5 H, s, H-8), 8.00 (0.3 H, s, H-8), 7.95 (0.2 H, s, H-2), 7.94 (0.3 H, s, H-2), 7.86 (0.3 H, s, H-2), 7.85 (0.2 H, s, H-2), 7.70 and 7.69 (total 0.5 H, each d, *J* = 8.6 Hz, J = 8.6 Hz, DNP), 7.61 (0.3 H, d, *J* = 8.6 Hz, DNP), 7.59 (0.2 H, d, *J* = 8.6 Hz, DNP), 7.42–7.41 (2 H, m, DMTr), 7.32–7.19 (7 H, m, DMTr), 6.83–6.78 (4 H, m, DMTr), 6.46–6.43 (0.5 H, m, H-1′), 6.41–6.38 (0.5 H, m, H-1′), 4.85–4.74 (1 H, m, H-3′), 4.36–4.34 (0.5 H, m, H-4′), 4.33–4.30 (0.5 H, m, H-4′), 3.91–3.83 (1 H, m, CH_2_), 3.78, 3.78, 3.77, 3.77, 3.77 (6 H, 5 s, OMe), 3.76–3.56 (3 H, m, CH_2_), 3.43–3.33 (2 H, m, H-5′a, H-5′b), 2.97–2.88 (0.5H, m, H-2′a), 2.83–2.60 (2.5 H, m, H-2′a, H-2′b, CH), 2.49–2.46 (1 H, m, CH), 1.21, 1.21, 1.20, 1.20, 1.19, 1.19, 1.18, 1.14, 1.12 (total 12 H, each s, CH_3_); ^13^C{^1^H} NMR (CDCl_3_, 125 MHz) δ 158.6, 158.6, 158.5, 158.5, 158.5, 155.2, 155.2, 148.1, 148.1, 147.4, 147.3, 147.2, 147.2, 146.4, 144.8, 144.6, 144.6, 144.5, 144.5, 144.4, 139.7, 139.5, 138.9, 138.8, 135.8, 135.7, 135.7, 135.6, 135.6, 135.6, 135.4, 135.4, 131.9, 130.2, 130.1, 130.1, 130.0, 128.8, 128.7, 128.2, 128.1, 128.1, 128.1, 127.9, 127.9, 127.0, 127.0, 126.9, 124.7, 124.6, 124.4, 124.3, 121.3, 117.6, 117.6, 117.5, 117.4, 113.2, 113.1, 113.1, 86.6, 86.6, 86.5, 86.5, 86.3, 86.2, 86.2, 86.0, 86.0, 85.9, 85.1, 85.1, 84.5, 84.5, 74.1, 73.9, 73.6, 73.5, 73.5, 73.4, 63.5, 63.4, 63.3, 63.2, 58.4, 58.3, 58.3, 58.2, 58.2, 58.1, 58.1, 57.9, 55.3, 55.2, 43.3, 43.3, 43.2, 43.2, 40.3, 40.3, 40.2, 40.2, 40.2, 39.9, 39.7, 39.7, 24.7, 24.6, 24.6, 24.6, 24.5, 21.5, 20.5, 20.4, 20.3, 20.2; ^31^P NMR (CDCl_3_, 202 MHz) δ 149.47, 149.37, 149.24.

### 4.3. Oligonucleotide Synthesis

CPG-supported ODNs were prepared on an H-6 DNA/RNA synthesizer (Nihon Techno Service, Ibaragi, Japan) using the corresponding phosphoramidite units (Glen research, Sterling, VA, USA) and CPG (Glen research) resin at a 0.4 μmol scale, according to the following procedure: detritylation (3% TCA in CH_2_Cl_2_, 70 s), coupling [0.25 M 5-benzyl-1*H*-tetrazole in dry acetonitrile, 12 min for 4 in 0.1 M dry acetonitrile, 30 s for natural nucleoside phosphoramidites in 0.065 M dry acetonitrile], capping [Ac_2_O in THF/pyridine and 1-methylimidazole in THF, 60 s], and oxidation [0.02 M I_2_ in THF/H_2_O/pyridine, 180 s]. After completion of synthesis, the CPG support was treated with ethylenediamine and/or 1,3-propanediamine for 30 min at 50 °C. The reaction was concentrated in vacuo, and the residue containing DMTr-ON products was combined with 0.2 M TEAA buffer (1.0 mL) and applied to a C18 cartridge column (YMC Dispo SPE C18, YMC, Kyoto, Japan). The cartridge was washed with 10% acetonitrile in 0.2 M TEAA (pH 7.0) to rinse failed sequences from the cartridge, and then with 2% trifluoroacetic acid to remove the DMTr group at the 5′-end. ODNs were eluted with 20–50% acetonitrile and evaporated to dryness before characterization by ESI-TOF mass analysis (see Appendix A). 

### 4.4. Calculation Details

Density functional theory (DFT) calculations on the model substrate were performed with the Gaussian09 suite of programs (Revision B.01, Gaussian, Inc., 340 Quinnipiac St., Bldg. 40, Wallingford CT 06492, USA) [24]. Geometries of all molecules and transition states were fully optimized without any symmetry constrains using the B3LYP method combined with the 6-31G*(d,p) basis set. Vibrational analyses were performed at the same level of theory on all optimized geometries to ensure that the optimized structures corresponded to local minima. All transition states were confirmed by one imaginary frequency. All energies reported in this paper and used for discussion refer to the sum of electronic and zero-point energies in Hartrees and relative energies in kcal/mol.

### 4.5. Thermal and Thermodynamic Analysis

Ultraviolet (UV) absorbance was measured on a UV-1800 spectrophotometer equipped with a temperature controller (SHIMAZU, Kyoto, Japan). Melting curves of DNA duplexes were acquired at 260 nm (*T*_m_ analysis, Figure 3) or 280 nm (stacking analysis, Figure 4) in 100 mM (*T*_m_ analysis, Figure 3) or 1.0 M (stacking analysis, Figure 4) NaCl, 10 mM Na_2_HPO_4_, and 1 mM Na_2_EDTA (pH 7.0). Samples were heated from 20 to 95 °C at a rate of 0.5 °C min^–1^. Before the measurements, the DNA duplexes were heated to 95 °C and then cooled to 20 °C. All melting curves were fitted to a theoretical equation to obtain thermodynamic parameters for double helix formation (∆*H*°, ∆*S*°, and ∆*G*°_37_) as described elsewhere [25]. We also evaluated these thermodynamic parameters from plots of the reciprocal of melting temperature (*T*_m_^−1^) versus ln(Ct/4). From the slope and intercept of the plots, thermodynamic parameters were obtained as described elsewhere [24]. We measured melting curves for at least 10 different concentrations of DNA duplex (1.0 μM to 40 μM for *T*_m_ analysis, Figure 3; 1.5 μM to 100 μM for stacking analysis, Figure 4). The thermodynamic parameters listed in Figure 3 and Figure 4 are the average values obtained from curve fitting, and plots of *T*_m_^−1^ versus ln (Ct/4).

### 4.6. Single Nucleotide Insertion Analysis

A primer labeled with FITC at the 5′-end (20-mer, 5′-FITC-GTTCTGGATGGTCAGCGCAC-3′) was annealed with a template (30-mer, ODN6; see Appendix A) in 10 mM Tris-HCl (pH 7.9) buffer containing 50 mM NaCl, 10 mM MgCl_2_, and 1 mM DTT. The primer–template duplex solution (final 0.2 or 0.8 μM) was mixed with each dNTP solution (final 0.05–5 μM). Each mixture was incubated for more than 2 min, and then the reaction was initiated by adding enzyme (final 0.025 units μL^−1^) to each duplex–dYTP mixture at 37  °C. Reactions were quenched with an equal amount of stop solution (0.1 % (*w*/*v*) bromophenol blue (BPB), 10  M urea, and 50 mM ethylenediaminetetraacetic acid (EDTA)). The diluted products were resolved by electrophoresis on a 20% polyacrylamide gel containing 8 M urea, and the gels were visualized with a Typhoon FLA 9500 scanner (Cytiva, Tokyo, Japan) equipped with ImageQuant^TM^ software (cytiva). Relative velocities (v0) were calculated as the extent of the reaction divided by the reaction time and were normalized to the duplex and enzyme concentration (0.2 μM, 0.025 units μL^−1^) for the various concentrations used. The kinetic parameters (*K*_m_ and *V*_max_) were obtained from Lineweaver–Burk plots of 1/v0 versus 1/[dNTP].

### 4.7. PCR and Sequencing Analysis

PCR was performed using Taq DNA Polymerase (New England Biolabs, Ipswich, MA, USA) with dNTPs. The reaction mixture contained 0.5 pM template, 1×ThermoPol^®^ Buffer (New England Biolabs), 0.5 μM primers (Forward, 5′-TAATACGACTCACTATAGGGACTAGCTACGAGTGCTC-3′; Reversed, 5′-GACGGAATATAAGCTGGTGG-3′), 0.2 mM dNTPs, and 0.025 units μL^−1^ of Taq DNA polymerase in a volume of 50 μL. The cycling conditions were 30 cycles of denaturation at 95 °C for 30 s, annealing at 55 °C for 30 s, and extension at 68 °C for 30 s. The reactions were analyzed by 6.4% polyacrylamide gel electrophoresis and stained with ethidium bromide. The amplicons were purified using a High Pure PCR Product Purification Kit (Roche Diagonostics, Tokyo, Japan), and the resulting products (1.125 μg) were each sequenced with 4 pmol of primer (5′-TAATACGACTCACTATAGGGACTAGCTACGAGTGCTC-3′) using an ABI PRISM 3100 Genetic Analyzer (Waltham, MA, USA).

## Data Availability

All data from this study are reported in the text or in “Appendix A”.

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
