# Peer review of "Synthesis and Behavior of DNA Oligomers Containing the Ambiguous Z-Nucleobase 5-Aminoimidazole-4-carboxamide"

_molecules, 2023, doi:10.3390/molecules28073265_

Round 1

Reviewer 1 Report

The authors have been able to devise a procedure for the synthesis of DNA oligomers containing the Z-base, 5-aminoimidazole-4-carboxamide, exploiting a "special" ring-opening reaction. Due to the rotatable nature of the carboxamide group, the Z-base could assume both an A-like configuration and a G-like configuration. The authors, thanks to a large series of UV-thermal denaturation experiments of suitably designed duplexes, have been able to conclude that the Z-base preferentially adopts an A-like configuration and forms A-T-like pairs. This base pairing ability has been confirmed by enzymatic approaches. In particular, in PCR experiments, Taq DNA polymerase incorporated dTTP as the nucleotide complementary to the Z-base in the template strand. 

I think the manuscript merits publication.

Reviewer 2 Report

In this manuscript, Nogi and et.al. developed chemical synthesis method to generate oligo nucleotides containing Z-base, used DFT calculation to explain the mechanism and side product, and used DNA polymerase and sequencing method to show that the Z-base acts as an A-like analog. I found the manuscript is very high quality, the experiment designs look great, and the data support the conclusion well.

Minor issues:

Line 21: as “an” A-like …

Line 124: spell outTEA and HFIP for the fist time

Line 232: Figure 3 f) legend is missing
